# The Performance Improvement of VLC-OFDM System Based on Reservoir Computing

Bingyao Cao, Kechen Yuan *, Hu Li, Shuaihang Duan, Yuwen Li and Yuanjiang Ouyang

Key Laboratory of Specialty Fiber Optics and Optical Access Networks, Shanghai Institute for Advanced Communication and Data Science, Shanghai University, Shanghai 200444, China; caobingyao@shu.edu.cn (B.C.); lihu@shu.edu.cn (H.L.); yizhi2019@shu.edu.cn (S.D.); li_yuwen@shu.edu.cn (Y.L.); oy_yuanjiang@shu.edu.cn (Y.O.)
* Correspondence: neymar@shu.edu.cn

**Abstract:** Nonlinear effects have been restricting the development of high-speed visible light communication (VLC) systems. Neural network (NN) has become an effective means to alleviate the nonlinearity of a VLC system due to its powerful ability to fit complicated functions. However, the complex training process of traditional NN limits its application in high-speed VLC. Without performance penalty, reservoir computing (RC) simplifies the training process of NN by training only part of the network connection weights, and has become an alternative scheme to NN. For the indoor visible light orthogonal frequency division multiplexing (VLC-OFDM) system, this paper studies the signal recovery effect of the pilot-assisted reservoir computing (PA-RC) frequency domain equalization algorithm. The pilot information is added to the feature engineering of RC to improve the accuracy of channel estimation by traditional least squares (LS) algorithm. The performance of 64 quadrature amplitude modulation (QAM) signal under different transmission rates and peak to peak voltage (Vpp) conditions is demonstrated in the experiments. Compared with the traditional frequency domain equalization algorithms, PA-RC can further expand the Vpp range that meets the 7% hard-decision forward error correction (FEC) limit of $3.8 \times 10^{-3}$. At the rate of 240 Mbps, the BER of the system is reduced by about 90%, and the utilization rate of the available frequency band of the system reaches 100%. The results show that PA-RC can effectively improve the transmission performance of VLC system well, and has strong generalization ability.

**Keywords:** visible light communication; orthogonal frequency division multiplexing; nonlinear equalization; reservoir computing



## 1. Introduction

It is pointed out in the potential key technologies of 6G that VLC, as a new spectrum resource technology, will become an important research direction in the future communication field [1,2]. VLC uses light-emitting diodes (LEDs) as light sources, and transmits data information with visible light while simultaneously illuminating. This means VLC has advantages such as rich spectrum resources, low cost, high transmission rate, and strong security [3]. In order to further improve the transmission rate of VLC system, relevant researchers have made a breakthrough in the research of high-order modulation format signals and advanced modulation technologies [4–7]. Among them, Orthogonal Frequency Division Multiplexing (OFDM) technology has become an important modulation technology to realize indoor high-speed VLC systems due to its advantages of high spectrum utilization and strong anti-multipath effect [8]. However, the limited modulation bandwidth and linear operating range of LEDs restrict the further development of VLC systems. The narrow modulation bandwidth causes the transmission signal on the high frequency subcarrier to suffer severe channel impairment [9]. With the increase of signal modulation order and communication rate, the nonlinear characteristics of LED will have

a more significant impact on the BER performance of VLC systems [10,11]. In addition, the inherent high peak to average power ratio (PAPR) of OFDM system makes the system more sensitive to the nonlinear effect of LED [12].

In order to alleviate the distortion of the received signal caused by the nonlinearity of VLC system, researchers mainly correct the distorted signal from the perspectives of pre- and post-distortion [13–17]. Compared with the pre-distortion algorithms, the post-distortion compensation algorithms have better performance by mitigating nonlinear interference existing in the entire communication system and transmission channel [18,19]. Post-distortion methods based on polynomial models usually require high model accuracy. The polynomial coefficients are difficult to determine in strong nonlinear and complex communication scenarios [20]. Traditional machine learning (ML) algorithms such as K-means clustering [21], support vector machine (SVM) [22], and K nearest neighbors (KNN) [23] generate nonlinear decision boundaries by learning the received data, which suppresses the influence of constellation distortion. Gaussian Mixture Model (GMM) divides data into several probability models based on Gaussian distribution, overcoming the limitation of Kmeans, taking distance as the only reference [24]. However, the above methods usually perform decision optimization on the equalized signal constellation. When the signal constellation diagram is seriously distorted, the traditional ML algorithms will not be able to achieve the expected performance.

With rapid development of integrated circuits and constant improvement of computer's computing power, deep neural network (DNN) with stronger fitting ability of nonlinear function has become an important method to improve the transmission performance of VLC systems. Among them, a series of DNN models, such as convolutional neural network (CNN) [25], dual-branch multilayer perceptron (MLP) [26] and long short-term memory network (LSTM) [27] with memory ability, are applied in VLC systems to overcome nonlinear effects. A deep learning model based on end-to-end optimization is proposed in [28], and the damage caused by each link in the VLC-OFDM system is reduced by introducing neural network models in different processing steps of the system. Ref. [29] takes channel impairment in the VLC-OFDM system as a learning task, and recovers distorted signals by introducing two DNN networks in the channel equalization and signal decision steps. In order to reduce the complexity of DNN, a dual network structure based on constellation decision is proposed in [30] to equalize the inner and outer ring signals of the constellation diagram, respectively. However, complex network structures are usually accompanied by higher training costs. A large number of parameters make the network converge slowly and easily fall into local minimum, resulting in poor generalization ability of the model. Reservoir Computing (RC) has recently become an alternative to traditional NN due to its excellent nonlinear processing capability and low training complexity [31–33]. RC, also known as Echo State Network (ESN), replaces the hidden layers of NN with a large-scale sparsely randomly connected network (reservoir). By training partial weights of the network, the training process of the algorithm is greatly simplified, and the shortcomings of the traditional recurrent neural network (RNN) structure are difficult to determine and the training process is too complicated. As a new type of random weight recursive network, it has been proved that RC can reduce the training complexity of NN and improve the performance of optical communication system [34].

In this paper, a pilot-assisted RC (PA-RC) nonlinear equalization algorithm is proposed to improve the transmission performance of the system. The shortcomings of the traditional least squares (LS) algorithm are improved by introducing PA-RC into the channel estimation stage. We analyze in detail several key parameters that affect the performance of the algorithm. The trend of bit error rate (BER) curve of 64QAM signal at different rates and different signal Vpp in the VLC-OFDM system and the performance improvement brought by the algorithm are experimentally studied. The results indicate that compared with zero forcing equalization (ZFE) algorithm, PA-RC can significantly improve BER performance of the system. At the rate of 240 Mbps, the Vpp range of the signal meeting the FEC threshold is expanded by more than 0.5 V. When the signal Vpp is 1.2 V, the system BER is reduced by 90%, and the available frequency band utilization rate reaches 100%. In addition, we

have proved through experiments that PA-RC has strong generalization ability, and the model trained under the condition of fixed signal-to-noise ratio (SNR) can be applied to the system under different transmission conditions. To the best of our knowledge, this is the first time that RC is used for equalization in VLC system.

## 2. The Proposed Scheme

### 2.1. Nonlinear Equalizer Based on Reservoir Computing

The PA-RC structure adopted in this paper is shown in Figure 1, which mainly includes the input layer, reservoir, and output layer. The input vector consists of valid data and received pilots within a frame. In a VLC-OFDM system, the received pilot data directly reflects the real situation of the channel. Due to the non-ideal characteristics of transmission system and channel, these pilots are inevitably subject to nonlinear interference. Therefore, the traditional LS channel estimation algorithm has high sensitivity to noise. In the proposed algorithm, the received pilot data is used as one of the features to learn the mapping relationship between input and output, thereby further improving the accuracy of channel estimation. The middle layer of the network uses a random sparsely connected network (reservoir) to replace the hidden layer of the traditional NN. The random connection between neurons makes the reservoir have short-term memory capacity. In this paper, the leaky integral neurons are used to replace ordinary neurons to optimize the performance of the RC algorithm. The output layer uses a linear activation function to linearly combine the neurons in the reservoir to output the recovered signal. The state and output equations of the proposed algorithm are shown in Equations (1) and (2):

$$S(n+1) = (1-a)S(n) + f(W_{in}X(n+1) + WS(n) + W_{back}Y(n)) \tag{1}$$

$$Y_{n+1} = f_{out}(W_{out}(x_{n+1}, S_{n+1}, Y_n)) \tag{2}$$

where $W_{in}$, $W$ and $W_{back}$ represent the input, the state variables, and output connection weight matrices to the state variables, respectively; $X(n) = [x_1(n), x_2(n), ..., x_K(n)]^T$ is the input vector at time $n$; $Y(n) = [y_1(n), y_2(n), ..., y_L(n)]^T$ is the output vector at time $n$; $S(n) = [s_1(n), s_2(n), ..., s_N(n)]^T$ is the state of N internal neurons at time $n$; $W_{out}$ represents the connection weight matrix of the reservoir, input, and output to output; $a$ is the leakage rate; $f$ and $f_{out}$ represent two different activation functions.

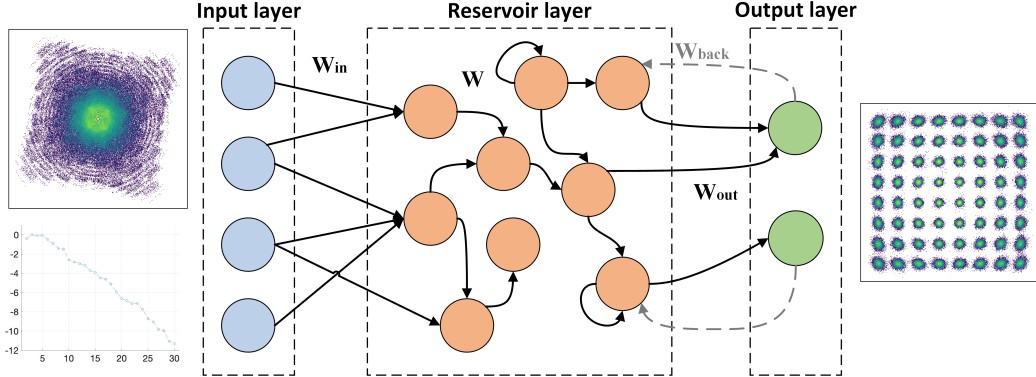

**Figure 1.** Diagram of PA-RC nonlinear equalizer.

In order not to lose phase information, the length of the input vector in the network is 4 (K = 4), which mainly includes the real and imaginary parts of the received data and pilot information. The output is the recovered signal constellation points, so the size of L is 2. The size of the reservoir is discussed in the experimental section. Equation (1) indicates that the state of the reservoir at a certain time is determined by the input vector at the current time, the state of the reservoir, and the output vector at the previous time. $a$ controls the retention of the reservoir's state at the last moment, which determines the long-term memory capability of the network. When $a$ is 1, the model degenerates back to

the ordinary ESN. *f* usually chooses *tanh* as the activation function to improve the nonlinear representation ability of the network. The state matrix $W$ represents the richness of the internal network of the reservoir. In order to reduce the computational complexity as much as possible, the sparsity of $W$ is generally maintained at 1% to 5%. Equation (2) shows that we need to train $W_{out}$ according to the input and output data of the system to obtain an output that is closer to the expected value. For the signal equalization problem, $f_{out}$ can take the linear identity function.

### 2.2. Training Process

Generally, the training process of RC includes two stages: states selection and weights calculation. In the state selection stage, the initial state of the reservoir needs to be determined first, and the initial state is usually assumed to be 0. The update process of the state is shown in formula (1). $W_{in}$, $W$, and $W_{back}$ are randomly generated and kept constant throughout the training phase. For simplicity, $W_{back}$ is assumed to be 0. The input vector and the state vector are spliced in a column to form a one-dimensional column vector $U$, and the $U$ obtained during the sampling period is formed into a state matrix $B$ by columns:

$$B = [U(m), U(m+1), ..., U(M)] \tag{3}$$

where $U(n) = [x_1(n), x_2(n), ..., x_K(n), s_1(n), s_2(n), ..., s_N(n)]^T$; $m$ is the sampling time; $M$ is the size of the training set.

In the weights calculation stage, the matrix $W_{out}$ needs to be calculated according to the collected system state matrix $B$ and training data, so that the actual output of the network is close to the expected output result. As shown in formula (4):

$$Y_{expect} \approx Y(n) = W_{out}^T B \tag{4}$$

where $Y_{expect}$ is the expected output.

To make the error between $Y(n)$ and $Y_{expect}$ as small as possible, we define the mean squared error (MSE) as the target loss function in the training phase and solve this linear regression problem using the least squares method. Computationally, this problem can be further processed as a pseudo-inverse problem of $B$:

$$W_{out} = Y_{expect} B^T (BB^T + \zeta I)^{-1} \tag{5}$$

Since $B$ may be ill-conditioned in practical applications, a pseudo-inverse algorithm or a regularization technique needs to be used to ensure that the above problems are solved. In Equation (5), we use ridge regression to process $B$. Improve the stability and reliability of traditional LS by adding a regularization term ($\zeta$). The completion of $W_{out}$ calculation represents the end of RC training process.

### 3. Experimental Setup

The VLC-OFDM transmission system based on 64QAM signals is shown in Figure 2. In order to ensure high frequency band utilization, DC-biased Optical Orthogonal Frequency Division Multiplexing (DCO-OFDM) modulation technology is adopted to generate positive real signals. Figure 2a shows the experimental setup of the indoor VLC system. At the transmitter, the offline random sequence is generated by MATLAB. The input data generates parallel data streams with the same number of sub-carriers by operations such as serial-to-parallel conversion, QAM mapping, and Hermitian symmetry. In order to avoid the influence of DC signal, the first subcarrier is set to 0. Before inverse fast Fourier transform (IFFT), a certain length of training sequence is added to the data frame. The training sequence is also called pilot data, and with the pilot information, the received signals can be synchronized and the channel response can be estimated in real time. In order to mitigate the inter-symbol interference (ISI) caused by the multipath effect, a cyclic prefix of a certain length needs to be added before the OFDM signal. By adding the pilot signal and

the cyclic prefix, a complete OFDM signal is generated. The time domain signals after IFFT and clipping are stored in the random access memory (RAM) of the field programmable gate array (FPGA). After that, the transmitted signal is converted into an analog signal by a digital-to-analog converter (DAC-AD9708), and positive real signals are generated by an adjustable attenuator and an amplifier (AMP-OPA657) followed by a DC offset, thereby Drive blue LED (LSLED405-5) with wavelength of 405 nm for electro-optical conversion to send optical signals into 1.2 m indoor free space.

A lens is placed at the receiver of the system to collect the beam to improve the SNR of the receiver. The collected received light signals are converted into electrical signals by a photodetector (Hamamatsu-S10784). Signals are then captured by a digital phosphor oscilloscope (Tektronix 7354C) at a rate of 10× upsampling. In the signal off-line processing stage, the received signals are synchronized, and parallel-serial conversion and fast Fourier transform (FFT) are used to obtain the frequency-domain OFDM data. The pilot data and valid data information are sent in a frame into the PA-RC network to estimate the channel and recover the distorted signal. Finally, the BER performance of the received data demodulated by QAM is analyzed. The spectrum of the transmitter and receiver of the system are shown in Figure 2b,c.

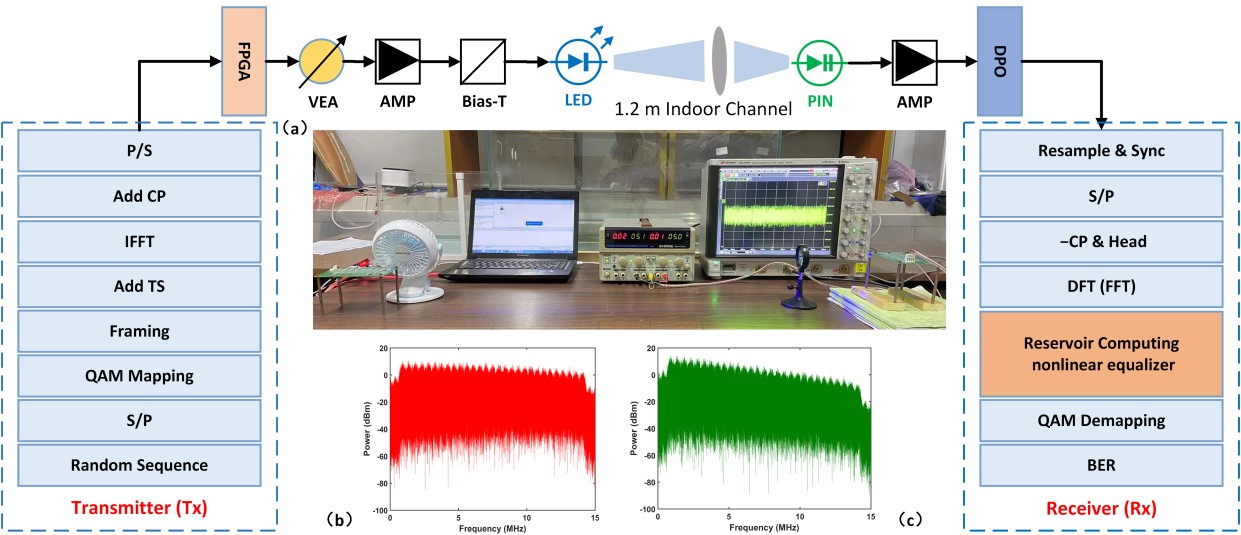

**Figure 2.** System structure diagram: (**a**) The experimental bench; (**b**) the transmitted spectra of the transmitter; (**c**) the received spectra of the receiver.

## 4. Experimental Results and Analysis

The OFDM related parameters used in the experiment are shown in Table 1. We use DCO-OFDM modulation technology to generate positive real signals that can drive LEDs to work properly. Therefore, the valid data sub-carriers are half of the total number of subcarriers. The performance of the first subcarrier is often poor because it is close to DC. Therefore, the first sub-carrier does not transmit valid data. A frame transmission sequence includes 100 OFDM symbols, the first of which is a training symbol. The transmitted sequences are generated completely randomly, and the training and test sets are 8000 and 10,000 in size, respectively. To demonstrate the adaptability to environmental changes of the model, training and test sets are collected at two different time periods.

We first discuss several important parameters related to the performance of PA-RC, namely the number of neurons in the reservoir, sampling time, spectral radius, leakage rate, scaling factor and sparsity, and take the system BER as the index to measure the performance of the algorithm. During the experiment, the default values of the above six parameters are 100, 1000, 0.1, 0.9, 0.1, and 0.3, respectively. We verify the effect of parameter changes on the algorithm performance under the condition that the signal Vpp is 1.2 V. In order to follow the principle of the control variable method, when a parameter is changed,

other fixed parameters use default values. A set of parameters with stronger generalization ability is selected by comparing the BER under different system transmission rates.

**Table 1.** Parameters of OFDM.

| Parameter | Value |
| --- | --- |
| Number of data-carrying subcarriers | 31 |
| FFT size | 64 |
| Length of cyclic prefix (CP) | 16 |
| Number of training symbols | 1 |
| Number of subcarriers near DC | 1 |
| QAM signal modulation order | 64 |
| Length of one frame | 100 |

The number of neurons in PA-RC determines the size of the reservoir. Figure 3a shows that PA-RC performance increases with the increase of the reservoir and gradually tends to be stable. It is not difficult to understand that when the number of neurons in the reservoir is small, it will lead to insufficient representation ability of the model for the nonlinear system. As the number of neurons increases, the network describes the system more accurately. When the number of neurons increases to a certain extent, the training complexity of the model increases, and the network may overfit. The function of the sampling time is to avoid the influence of the reservoir's initial state on the network performance. As shown in Figure 3b, on the nonlinear characterization problem of VLC system, a small sampling time can make the model achieve better performance. The performance of the model starts to deteriorate after the 4000th sample point. A later sampling time means a smaller number of reservoir states and expected outputs to be collected. The linear regression process may increase the training error of RC due to insufficient number of samples. Equation (5) indicates that the size of the state matrix $B$ directly determines the computational complexity of $W_{out}$, and the size of $B$ is determined by the number and sampling time of reservoir neurons. In order to balance model performance and training complexity, the number of neurons and sampling time are taken as 100 and 4000, respectively.

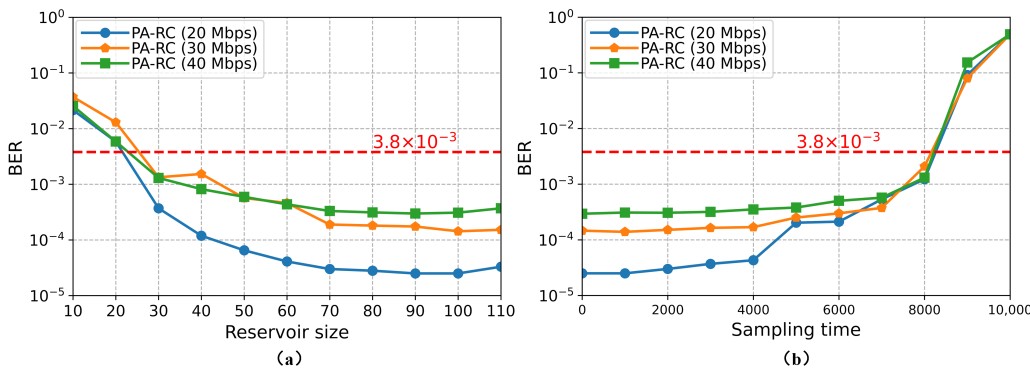

**Figure 3.** BER vs. RC parameters. (**a**) Neurons in the reservoir; (**b**) sampling time.

The spectral radius is defined as the absolute value of the largest eigenvalue of the connection weight matrix $W$, which is an important condition to ensure the stability of the algorithm. When spectral radius is less than 1, the network has the property of echo. As shown in Figure 4a, the system BER increases gradually with the increase of spectral radius. In the range of 0 to 0.2, the performance of the network is optimal. The proposed algorithm uses leaky integral neurons to replace ordinary neurons in the reservoir. Therefore, we discuss the impact of leakage rate on the performance of PA-RC. As shown in Figure 4b, a small value of leakage rate results in a slow response of the reservoir to the input signal. By increasing the leakage rate, the performance of the algorithm is significantly improved.

In the range of 0.9 to 1, the performance of PA-RC tends to be stable. When the leakage rate is 1, the leaky integral neurons degenerate into ordinary neurons. Theoretically, leaky integral neurons can improve the performance of the reservoir. Without loss of generality, the values of spectral radius and leakage rate are 0.1 and 0.98, respectively.

The scaling factor is mainly used to adjust the feature scale of the input signal. When the value of scaling factor is small, the network element works around the linear center of activation function. As the scaling factor increases, the internal units of the network gradually approach the saturation point of *f*, and the nonlinearity of the model is stronger. As shown in Figure 5a, a value between 1 and 2 is more appropriate for the scaling factor. Sparsity represents the proportion of non-zero elements to the total elements in the neuron connection matrix of the reservoir. Generally, the smaller the sparsity is, the simpler the internal structure of the reservoir is, but its nonlinear representation ability is weakened. From Figure 5b, we find that for the problem of the nonlinear equalization of the VLC system, the sparsity within a fixed range (1~5%) has little effect on the performance of the algorithm. In order to reduce the network complexity of the reservoir layer as much as possible, we set the sparsity as 0.01. Since then, we have identified all parameters that affect the computational performance of the reservoir. The parameters related to PA-RC are shown in Table 2.

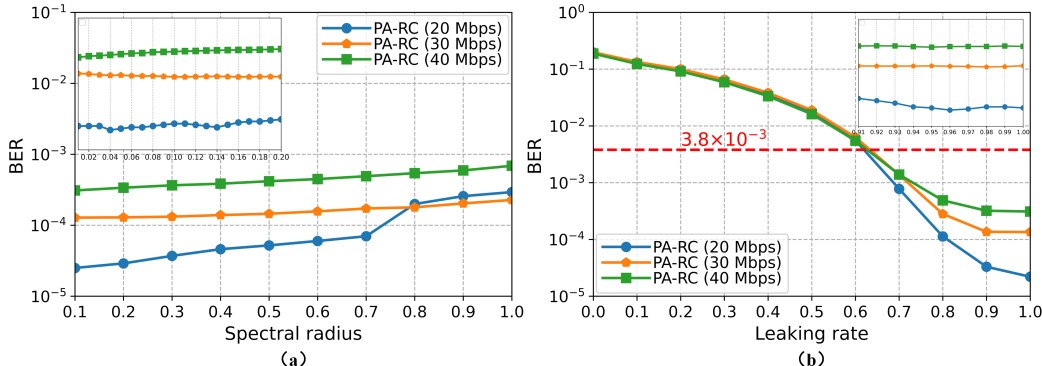

**Figure 4.** BER vs. RC parameters. (**a**) Spectral radius; (**b**) leaking rate.

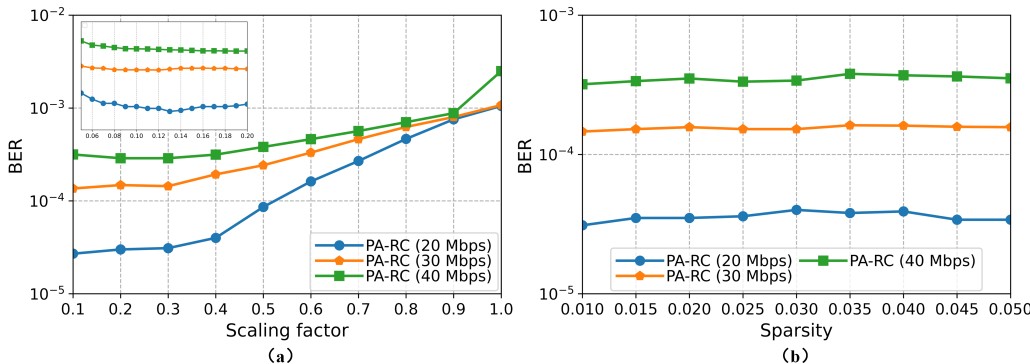

**Figure 5.** BER vs. parameters. (**a**) Scaling factor; (**b**) sparsity.

**Table 2.** Parameters of PA-RC.

| Parameter | Value |
| --- | --- |
| Number of neurons in the reservoir | 100 |
| Sampling time | 4000 |
| Spectral radius | 0.1 |
| Leakage rate | 0.98 |
| Scaling factor | 0.12 |
| Sparsity | 1% |
| Training set size | 8000 |
| Test set size | 10,000 |

We increase the nonlinearity of the system by changing the signal Vpp and compare the effect of system performance at different rates. The signal Vpp directly reflects the size of the current system SNR. The LS curve represents the system BER recovered by LS channel estimation combined with ZFE. The LS + RC curve represents the BER of the signal demapped by RC after ZFE. PA-RC represents the BER curve of the signal recovered by the algorithm used in this paper. As shown in Figure 6, it is obvious that under different transmission rates, the system BER shows a trend of first decreasing and then increasing. When Vpp is small, due to the low SNR of the system, Gaussian White Noise (GWN) interferes greatly with the transmitted signals. With the increase of Vpp, BER decreases. When Vpp increases to a certain extent, the system performance begins to deteriorate. This is easy to understand, because the linear operating range of non-linear devices such as LED is limited. When the signal power is high, the LED will enter the non-linear operating range and the signal will be clipped and distorted.

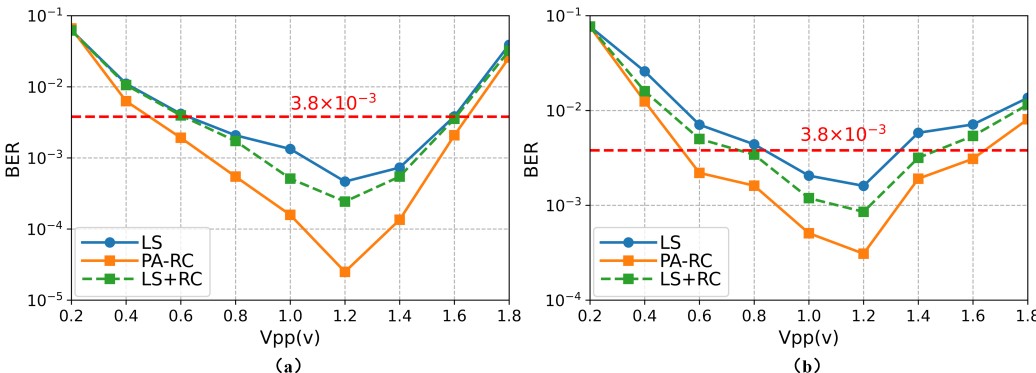

**Figure 6.** BER vs. signal Vpp. (**a**) The data is 120 Mbps; (**b**) the data is 240 Mbps.

As shown in Figure 6a, under the rate of 120 Mbps, with the help of the ZFE to equalize received signals, the Vpp range of the signal whose system BER meets the FEC threshold ($3.8 \times 10^{-3}$) is about 1.0 V. By using the RC algorithm to further restore the ZFE equalized signals, the system BER is reduced, but the range of Vpp is not expanded. because the recovery effect of ZFE completely depends on the accuracy of the channel estimation by LS. Due to the serious attenuation of the channel response of high-frequency subcarriers, the channel information estimated by the pilot is strongly interfered by noise. Therefore, RC cannot further recover the signals. PA-RC can better recover received signals by correcting the channel information estimated by the pilot. The BER is greatly reduced, and the Vpp range is expanded to about 1.2 V. When the Vpp is about 1.2 V, the system BER performance reaches the best, and the proposed algorithm reduces the system BER by more than 95%. When the system transmission rate increases to 240 Mbps, the BER performance deteriorates, as shown in Figure 6b. This is because as the data rate increases, the number of light emissions per unit time increases, resulting in enhanced nonlinearity. Using ZFE to recover the received signals, the Vpp range that satisfies the FEC threshold is about 0.5 V. After using RC to demap the signal based on the LS channel estimation, the Vpp range

of the signal satisfying the FEC threshold is expanded to 0.6 V. The influence of channel impairment is compensated by PA-RC, and the Vpp range of the signal satisfying FEC is expanded to about 1.1 V. From the results, the system transmission performance has been significantly improved. When the signal Vpp is 0.6 V and 1.6 V, the BER is still below the FEC threshold. It shows that RC can still bring some improvement to the system under the condition of low SNR and strong nonlinear interference. When Vpp is 1.2 V, the system BER is reduced by about 90%.

In order to reflect the improvement of the utilization rate of the available frequency band of the system brought by PA-RC, we draw the BER curves of high frequency subcarriers under different SNR. As shown in Figure 7, the BER of the subcarriers gradually deteriorates with the increase of frequency. The main reason is that the modulation bandwidth of nonlinear devices such as LED is limited, resulting in the gradual attenuation of VLC channel response with the increase of frequency. When Vpp is 0.6 V, the transmitted signals are seriously interfered by GWN, and the transmission performance of high-frequency subcarrier deteriorates seriously. As shown in Figure 7a, starting from the 23rd subcarrier, the BER is already above the FEC threshold. After using PA-RC to restore the received signals, the number of available sub-carriers reaches 28. When Vpp is 1.2 V, the number of available subcarriers in the system increases significantly under optimal transmission conditions. The PA-RC makes the utilization rate of the available frequency band of the system reach 100%. When Vpp is 1.5 V, the transmitted signals suffer severe clipping distortion. By using PA-RC to suppress nonlinear effects in the system, the available frequency band utilization of the system is improved by about 16%. This indicates that the proposed algorithm can further improve the transmission performance of high-frequency subcarriers without losing the system transmission capacity.

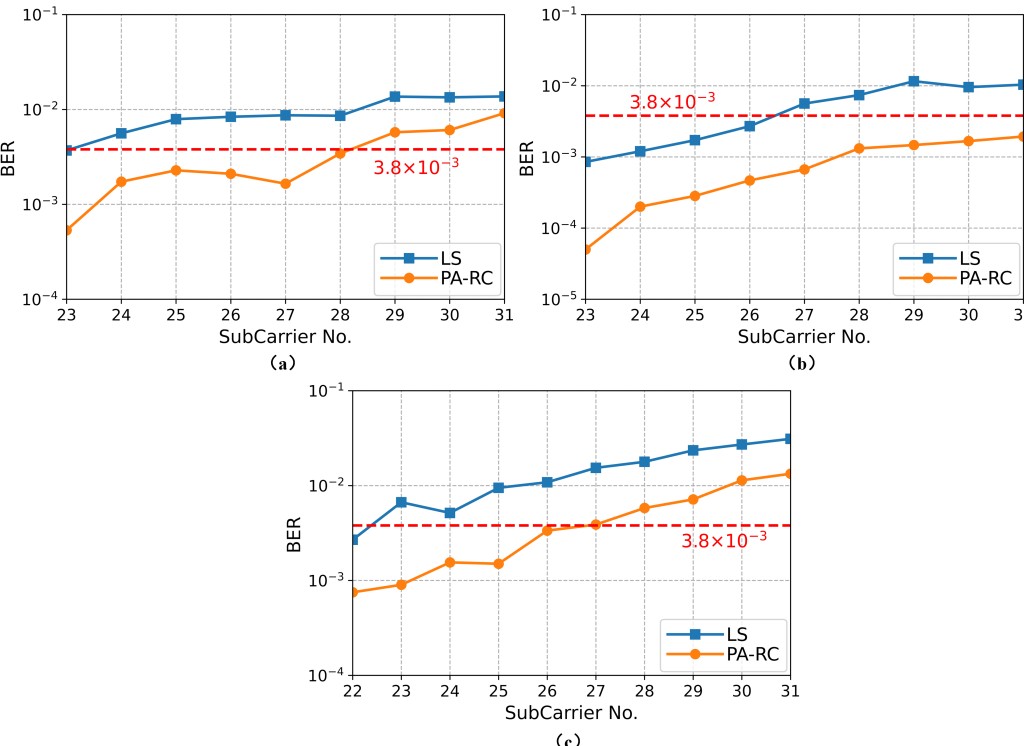

**Figure 7.** BER vs. Subcarrier number under the rate of 240 Mbps. (**a**) The Vpp is 0.6 V; (**b**) the Vpp is 1.2 V; (**c**) the Vpp is 1.5 V.

We verified the adaptability and sensitivity of PA-RC to changes in the system transmission environment at the transmission rate of 240 Mbps. The received signals under other SNR conditions are recovered by the model trained under the fixed SNR, and the system BER is used as the judgment index of the algorithm performance. As shown in

Figure 8, PA-RC (0.6 V), PA-RC (1.2 V), and PA-RC (1.5 V) represent the models trained with signal Vpp of 0.6 V, 1.2 V, and 1.5 V, respectively. PA-RC (0.6 V) has poor performance when Vpp is greater than 1 V. When Vpp is less than 1.2 V, the equalization performance of the PA-RC (1.5 V) is inferior to that of PA-RC (0.6 V) and PA-RC (1.2 V). It can be seen from the previous analysis that different Vpp conditions represent different interferences on the signals. Therefore, when the signal Vpp is quite different from the conditions during model training, the algorithm will not be able to achieve the best equalization performance. Note that PA-RC (1.2 V) has better performance under different transmission conditions, which indicates that the model trained around 1.2 V has the best generalization ability. Figure 8b shows the comparison of the BER performance of PA-RC (0.6 V) and LS under the optimal transmission conditions of the high-frequency sub-carriers of the system. Due to the great changes in transmission conditions, the performance of the PA-RC algorithm is degraded, but the recovery effect of the signal is still better than that of LS. The results in Figure 8 show that even the models trained under low SNR (0.6 V) and high SNR (1.5 V) conditions still have excellent equalization performance in changing environments.

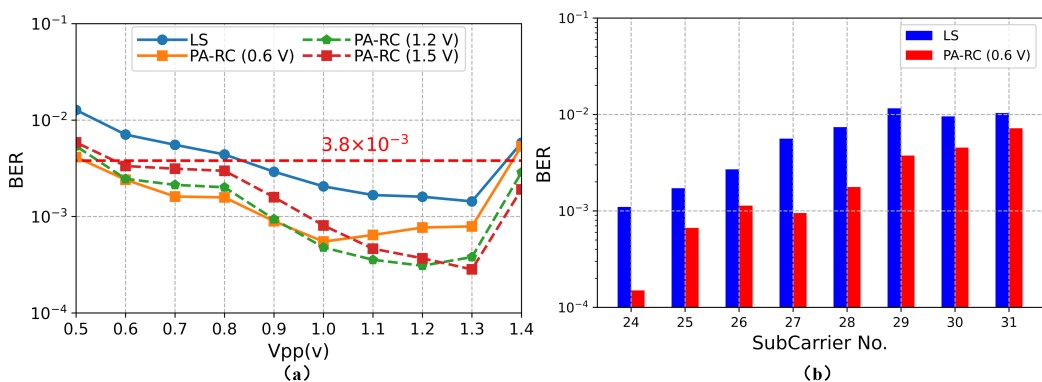

**Figure 8.** The data rate is 240 Mbps: (**a**) BER vs. Vpp; (**b**) BER vs. Subcarrier number when Vpp is 1.2 V.

In order to verify that RC has the same performance as the traditional NN while simplifying the model training, we choose two representative algorithms, the fully connected neural network (FCNN) and the long short-term memory network (LSTM) trained under different iterations, as the comparison objects. For the convenience of comparison, both FCNN and LSTM adopt a single hidden layer structure. By changing the number of neurons in the middle layer, the algorithm performance under different network scales can be compared. When Vpp is 1.2 V, as shown in Figure 9a, FCNN (10 epochs) and LSTM (10 epochs) have poor signal recovery performance. When N is 100, the two networks have only the same performance as LS ($1.92 \times 10^{-3}$). When the epochs is 30, the performance of FCNN and LSTM is close to RC. It shows that traditional NN needs more iterations to converge. However, PA-RC only needs to solve a simple linear regression problem during the training process, so the convergence rate is fast. We note that the PA-RC algorithm outperformed LS with 30 neurons, while FCNN (10 epochs) and LSTM (10 epochs) required 40 neurons. When Vpp is 1.5 V, the system suffers from strong nonlinear disturbance. As shown in Figure 9b, under the condition of fewer neurons, FCNN and LSTM improve the performance through more iterations. When N is 30, LSTM (30 epochs) and FCNN (30 epochs) outperform LS ($5.82 \times 10^{-3}$). With the increase of neurons in the middle layer, the PA-RC converges rapidly and achieves the same performance as the two NNs. The results in Figure 9 indicate that PA-RC has the same excellent nonlinear mitigation ability as traditional NN, and training is simple and the convergence rate is fast. Furthermore, we find that LSTM has similar performance to FCNN. Mainly because the objects of equalization in this paper are the frequency domain signals. Therefore, LSTM does not exert its advantages in time series problems. Compared with [29] using two NNs, We use PA-RC to directly equalize the received signals, which greatly reduces the complexity of the receiver.

In addition, the number of neurons in the reservoir of PA-RC is small, which is the most essential difference from the NN equalizer used in current VLC systems.

Finally, we show the received signal constellations on the 10th and 31st sub-carriers of OFDM symbols under transmission rate of 240 Mbps. Figure 10a,c show the restored constellations of the received signals on the 10th and 30th subcarriers by ZFE, respectively. We find that the signal constellations on the high frequency subcarriers produce obvious rotation. This indicates that there is phase noise in the system. At high frequencies, due to more serious channel impairments, the estimation of the channel by the traditional LS algorithm will become inaccurate, so the effect of phase noise is more obvious. Figure 10b,d show the recovery effect of PA-RC on the signals. Compared with ZFE, the signal distribution after PA-RC restoration is more dense. It indicates that the proposed algorithm has a better equalization performance on received signals. The channel information estimated by pilots is corrected by PA-RC, and the common phase error (CPE) of the received signals is effectively eliminated. We know that the constellation points of the 64QAM signal are distributed on seven circles of constant modulus length. It is not difficult to find in Figure 10 that the signals on the outer ring are affected by more serious nonlinearity. Mainly because the signals on the outer ring have higher power, in addition to amplifying the multiplicative noise in the system, they will make it easier for nonlinear devices such as LEDs to enter the nonlinear working area. The signals are corrected by PA-RC, and the CPE and amplitude noise of the high frequency part are effectively suppressed. However, there are still some phase noise that cannot be completely eliminated. In future work, we will combine some commonly used phase noise suppression algorithms to better recover the signals.

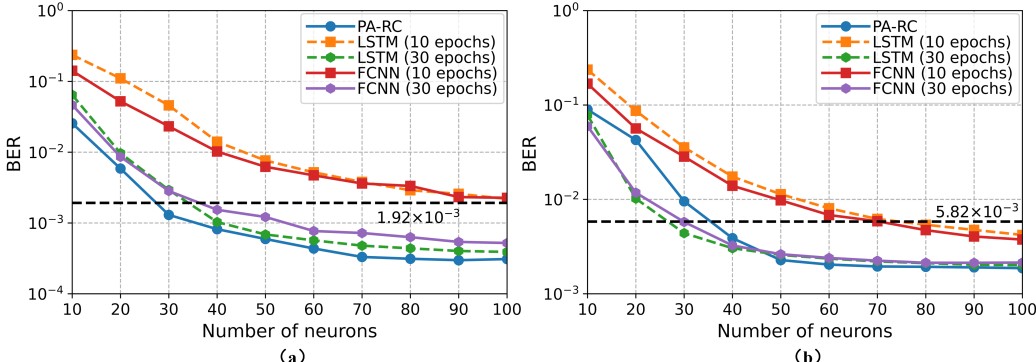

**Figure 9.** BER vs. Number of neurons. (**a**) The Vpp is 1.2 V; (**b**) the Vpp is 1.5 V.

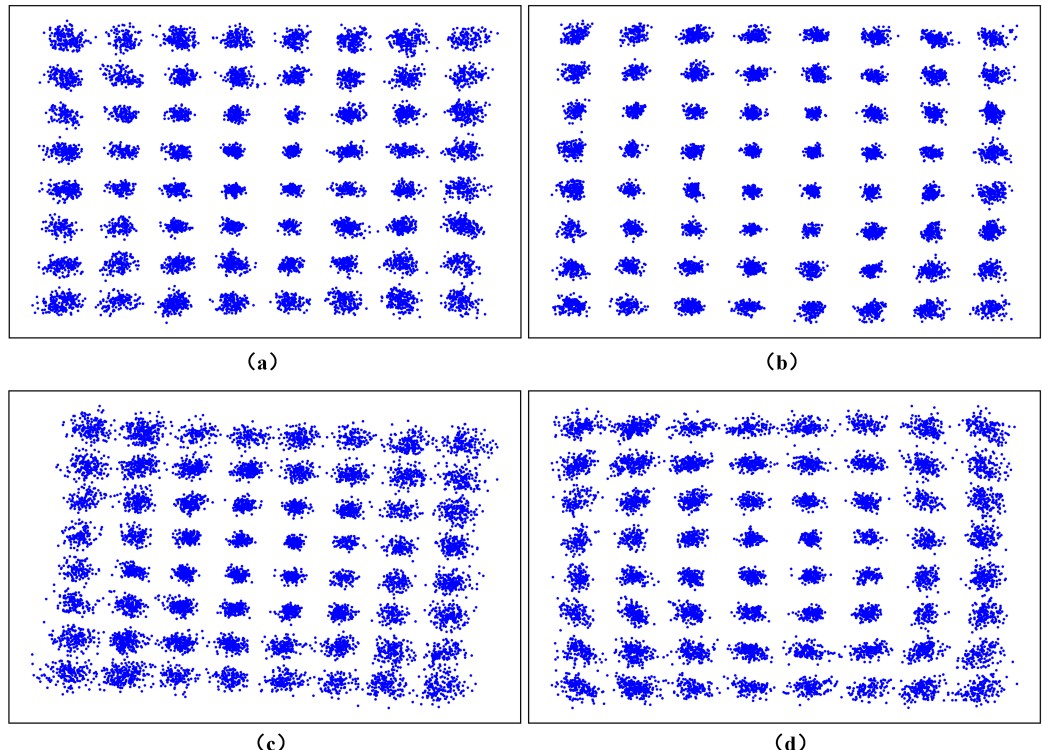

**Figure 10.** Constellation diagram of received signal on the 10th subcarrier recovered by (**a**) LS and (**b**) RC; constellation diagram of received signal on the 31th subcarrier recovered by (**c**) LS and (**d**) RC.

## 5. Conclusions

In this paper, we adopt PA-RC to compensate for the signals subjected to nonlinear interference in a VLC-OFDM system. Taking the received pilot information as one of the features of the input vector enables the model to obtain the true state of the current channel, thereby better recovering signals. The experiment compares the signal recovery effect of PA-RC and traditional frequency domain equalization algorithm under strong nonlinear effects conditions. We study the relationship between system BER performance and signal Vpp. When Vpp is too small or too large, the signals are severely distorted. The results show that PA-RC can significantly reduce the system BER. At the transmission rate of 120 Mbps, compared with ZFE, PA-RC expands the Vpp range by 0.2 V. When Vpp is 1.2 V, the system BER is reduced by more than 95%. As the transmission rate increases, the system BER performance drops significantly. When the transmission rate is 240 Mbps, even under the optimal transmission conditions, the high-frequency subcarriers still cannot meet the transmission conditions. By utilizing PA-RC, the range of signal Vpp meeting FEC threshold is expanded from 0.5 V to 1.1 V. When Vpp is 1.2 V, the utilization rate of the available frequency band of the system reaches 100%, and the system BER is reduced by about 90%. Furthermore, we investigate the effect of system environment changes on PA-RC performance. The trained PA-RC model has stronger generalization ability when Vpp is 1.2 V. In order to simplify the training process of the RC, the reservoir used in this paper is randomly generated. The optimal reservoir structure should match the specific problem. In the future, it is believed that more diverse RC network structures will be applied in the VLC system to further alleviate nonlinear effects.

**Author Contributions:** Conceptualization, B.C. and K.Y.; methodology, K.Y.; software, K.Y. and S.D.; validation, K.Y. and S.D.; formal analysis, B.C.; investigation, K.Y.; resources, B.C.; data curation, K.Y. and H.L.; writing—original draft preparation, B.C.; writing—review and editing, K.Y., Y.L. and Y.O.; visualization, K.Y. All authors have read and agreed to the published version of the manuscript.

**Funding:** This work was supported in part by National Key Research and Development Program of China (2021YFB2900800 2021YFB2900802), the Science and Technology Commission of Shanghai Municipality (Project No. 20511102400, 20ZR1420900) and 111 project (D20031).

**Institutional Review Board Statement:** Not applicable.

**Informed Consent Statement:** Informed consent was obtained from all subjects involved in the study.

**Conflicts of Interest:** The authors declare no conflict of interest.

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
