# Peer review of "The Performance Improvement of VLC-OFDM System Based on Reservoir Computing"

_photonics, doi:10.3390/photonics9030185_

Round 1

Reviewer 1 Report

The writing and presentation quality of this work is nice. Authors have explained results very well. In this study, authors have studied signal recovery effect of the pilot-assisted reservoir computing (PA-RC) frequency domain equalization algorithm. The results presented in this study state that PA-RC can effectively improve the transmission performance of VLC system well, and has strong generalization ability. Its an interesting article which proposes a new research direction to use RC for equalization in VLC system.  I don’t have any specific comments as authors have nicely presented this work and clearly explained each result. A few suggestions are

  1. Line 11, what is QAM? Quadrature Amplitude Modulation. Authors should check each abbreviation and define at first place of appearance so readers won’t face any difficulty.
  2. Typo in line 25, VLC have (has) the advantages…
  3. In the end of 1st paragraph of Introduction section, authors can add more discussion about LED non-linearity effect from relevant articles.
  4. Typo in line 41, so They (they) are …
  5. Line 146, output result. (spacing required)As shown in formula…
  6. In the state selection stage, why did author consider the common case by both assuming initial stage and Wback as 0?
  7. Why did you choose DCO-OFDM modulation? Any specific advantage?
  8. Authors have taken N and M values as 100 and 4000 (Figure 3) to balance model performance and training complexity. What will be the effect if we consider lower value such as M=2000?
  9. Authors have computed sparsity curve between 1-5% and said it has little effect on algorithm. Can you show 1-10% and let us see clear difference.
  10. Clear explanation of basic parameters and analysis is provided. The comparison with FCNN and LSTM is good.
  11. Line 344, we found that LSTM has similar performance to FDNN? What is FDNN? Is it FCNN?
  12. Authors have provided very recent and interesting relevant  articles in references.
  13. Authors can add any possible modification or future research direction in Conclusions for relevant research fraternity.

In my opinion, the overall research contribution is good and authors have discussed results very well. The manuscript is nicely written and organized.

Reviewer 2 Report

The authors utilized reservoir computing to solve the nonlinear effects in VLC-OFDM. The performance of 64QAM signal under different transmission rates and peak to peak voltage conditions is demonstrated in the experiments. The experiments results are reasonable and interesting. Some typos need attentions, e.g., line 41, "so They", "T" should be written as "t".  

Reviewer 3 Report

In this paper, the authors present use the signal recovery effect of the pilot-assisted reservoir computing frequency domain equalization algorithm. The pilot information is added to the feature engineering of RC to improve the accuracy of channel estimation by the traditional least-squares algorithm. The performance of the 64QAM signal under different transmission rates and peak to peak voltage conditions is demonstrated in the experiments. Which have some potential applications for the VLC-OFDM system. This article is clear, concise, and suitable for the scope of the journal. Several small suggestions are supplied:
1. Suggest increasing the length label for the inset point graph in Fig.1.
2. Suggest the authors give more detail about the results present here compare with results in the works of literature.
3. Suggest improving the draft with a native speaker.
